# UrbanWorld: An Urban World Model for 3D City Generation

## Abstract

Cities, as the essential environment of human life, encompass diverse physical elements such as buildings, roads and vegetation, which continuously interact with dynamic entities like people and vehicles. Crafting realistic, interactive 3D urban environments is essential for nurturing AGI systems and constructing AI agents capable of perceiving, decision-making, and acting like humans in real-world environments. However, creating high-fidelity 3D urban environments usually entails extensive manual labor from designers, involving intricate detailing and representation of complex urban elements. Therefore, accomplishing this automatically remains a longstanding challenge. Toward this problem, we propose UrbanWorld, the first generative urban world model that can automatically create a customized, realistic and interactive 3D urban world with flexible control conditions. Specifically, we design a progressive diffusion-based rendering method to produce 3D urban assets with high-quality textures. Moreover, we propose a specialized urban multimodal large language model (Urban MLLM) trained on realistic street-view image-text corpus to supervise and guide the generation process. UrbanWorld incorporates four key stages in the generation pipeline: flexible 3D layout generation from OSM data or urban layout with semantic and height maps, urban scene design with Urban MLLM, controllable urban asset rendering via progressive 3D diffusion, and MLLM-assisted scene refinement. We conduct extensive quantitative analysis on five visual metrics, demonstrating that Urban-World achieves state-of-the-art generation realism. Next, we provide qualitative results about the controllable generation capabilities of UrbanWorld using both textual and image-based prompts. Lastly, we verify the interactive nature of these environments by showcasing the agent perception and navigation within the created environments. We contribute UrbanWorld as an open-source tool available at https://github.com/Urban-World/UrbanWorld.

## 1 Introduction

Cities are the most complex human-centric environments, characterized by their intricate spatial structures, heterogeneous components such as buildings, infrastructure, and public spaces, and dynamic interactions between these components and human activities. Creating near-realistic 3D urban world environments is a fundamental technique for broad research and real applications across various domains such as Artificial General Intelligence (Zhang et al., 2024), AI agents (Yang et al., 2024), embodied AI (Wang et al., 2024a), urban simulation (Xu et al., 2023) and metaverse (Allam et al., 2022). Traditionally, achieving this involves expensive labor costs for human designers on detailed asset modeling, texture mapping, and scene composition. With the advancement of generative AI, there have emerged some automatic approaches for 3D scene generation based on volumetric rendering(Lin et al., 2023; Xie et al., 2024) and diffusion models(Deng et al., 2023; Lu et al., 2024). These approaches have revolutionized the paradigm of 3D scene generation, alleviating the high costs of manual design. However, the crafted 3D scenes are limited to the video format and unable to provide embodied and interactive environments. Regarding this issue, a recent series of methods known as world models have emerged, preliminarily focusing on autonomous driving scenes (Hu et al., 2023; Wang et al., 2023b). These models are shown to possess the capability of understanding the scene dynamics and predicting future states, uplifting the interactivity of 3D scene generation. However, there is still a large gap between the created urban environments and the real urban world

in which humans live. To sum up, there is still a long way from the actual "urban world models", which we define as models able to create urban environments that are (1) realistic and interactive (2) customizable and controllable (3) capable of supporting embodied agent learning. Table 1 provides a comprehensive review of existing 3D city generation methods from these key perspectives, highlighting that no existing approach can fully satisfy these requirements.

Urban world models are of great significance in developing embodied intelligence and Artificial General Intelligence (AGI). Firstly, it is promising to bridge the gap between virtual environments and the real world, enabling embodied agents to interact with and learn from richly detailed, realistic urban environments. Secondly, by crafting synthetic 3D urban environments, researchers can gain complete control over data generation, with full access to all generative parameters. Machine perceptual systems can thus be trained on tasks that are not well suited to conduct in the real world or require various environments. Finally, a sophisticated urban world model can simulate a wide variety of environments, from bustling city centers to quiet residential neighborhoods, with realistic visual appearances of physical infrastructures such as buildings, roads, and natural spaces. This is crucial to avoid overfitting and creating agents with high generalization in diverse and dynamic environments. However, there are no specialized urban world models for automatically crafting interactive 3D urban environments, hindering the advancement of AI abilities toward general intelligence through embodied learning in complex open-world environments.

Toward this issue, we propose UrbanWorld, a generative urban world model that can automatically create realistic, controllable and embodied 3D urban environments from user instructions and urban layout data in various format such as OpenStreetMap[1] (OSM) and layouts with semantic and depth maps (Deng et al., 2023; He & Aliaga, 2024). In detail, there are four key modules in the framework of UrbanWorld. Firstly, UrbanWorld automatically generates untextured 3D layouts with the above-mentioned input data and conducts detailed asset processing via Blender[2]. Then, UrbanWorld adopts a fine-tuned urban-specific multimodal large language model (called Urban MLLM) to effectively plan and design urban scenes following user instructions, generating detailed textual descriptions of urban elements. Next, UrbanWorld integrates a 3D asset renderer based on texture diffusion and refinement, flexibly controlled by textual and visual conditions. Finally, to further optimize the visual appearance, it utilizes Urban MLLM to scrutinize the crafted 3D urban environment, generating detailed suggestions for refinement and activate an additional iteration of rendering.

Our framework is highly flexible to support generating two typical types of 3D urban environments. On the one hand, UrbanWorld can generate a highly accurate replica of the real urban environment with real street-view imagery as the generation condition, which holds significant potential for studies of urban planning and geographic information systems. On the other hand, it can also generate fully customized urban environments with textual descriptions as the generation condition. This capability is valuable for simulating and exploring hypothetical urban scenarios, especially in areas such as virtual city design, gaming, and emergency response planning. In the experimental part, we begin by conducting a comprehensive quantitative evaluation using five visual metrics, validating the state-of-the-art realism of UrbanWorld's generated environments. We then showcase diverse generation results from various textual and image prompts, highlighting the superior controllability of UrbanWorld. Finally, we emphasize the interactive nature of the generated 3D urban environments by demonstrating agent perception and navigation within them. We contribute UrbanWorld as an open platform to support the creation and manipulation of more advanced 3D urban environments, facilitating the advancement of broad AI communities.

The contributions of this work can be summarized as follows:

- We present UrbanWorld, the first urban world model for automatically creating realistic, customized and interactive embodied 3D urban environments with flexible controls.

- UrbanWorld demonstrates its superior generative ability to craft high-fidelity 3D urban environments, greatly enhancing the authenticity of interactions in the environment.

- We provide UrbanWorld as an open-source platform to develop 3D urban environments, which can support a wide range of research including embodied intelligence and AI agents, further laying a foundation for the advancement of AGI.

---

[1]https://www.openstreetmap.org/
[2]https://www.blender.org/

Table 1: Comparison between existing works for 3D city generation and UrbanWorld from four aspects: text-controllable, image-controllable, whether new assets can be created, and interactive.

| Type | Method | Text-controllable | Image-controllable | Creating new assets | Interactive |
|---|---|---|---|---|---|
| Neural Rendering | SceneDreamer (Chen et al., 2023) | × | ✓ | ✓ | × |
| | PersistentNature (Chai et al., 2023) | × | ✓ | ✓ | × |
| | Infinicity Lin et al. (2023) | × | ✓ | ✓ | × |
| | CityDreamer (Xie et al., 2024) | × | × | ✓ | × |
| Diffusion | CityGen (Deng et al., 2023) | × | × | ✓ | × |
| 3D Modeling Software | MetaUrban (Wu et al., 2024a) | × | × | × | ✓ |
| | SceneCraft (Hu et al., 2024) | ✓ | × | × | ✓ |
| | CityCraft (Deng et al., 2024) | ✓ | × | × | ✓ |
| Comprehensive | UrbanWorld | ✓ | ✓ | ✓ | ✓ |

## 2 RELATED WORKS

### 2.1 3D URBAN SCENE GENERATION

3D urban scene generation aims to create realistic 3D urban environments with sophisticated urban planning and visual element design, usually requiring high human efforts such as complex asset modeling, texture mapping, and scene composition. With the advancement of deep learning techniques, recently there are three lines of work trying to achieve this in an automated way, including neural rendering-based methods (Lin et al., 2023; Xie et al., 2024; Chen et al., 2023), diffusion-based methods (Deng et al., 2023; Inoue et al., 2023; Wu et al., 2024b) and 3D modeling software-based methods (Zhou et al., 2024; Hu et al., 2024; Wu et al., 2024a). An overview of the method comparison is shown in Table 1. Neural rendering-based methods implicitly represent the urban scene and perform the volumetric rendering for the neural fields. For example, CityDreamer (Xie et al., 2024) first separates the scene into buildings and backgrounds then introduces different types of neural fields for asset rendering. These methods can produce a high-quality visual appearance while potentially losing geometric fidelity. Diffusion-based methods utilize diffusion models to generate city layouts or urban scenes. CityGen (Deng et al., 2023) provides an end-to-end pipeline to create diverse 3D city layouts with Stable Diffusion. These methods are creative in generating scene images or videos, but hard to obtain embodied 3D environments, limiting the practical usages. Recently, some professional software script-based methods have been proposed, trying to develop an automatic agentic workflow using LLMs to control the professional software for scene creation. CityCraft (Deng et al., 2024) adopts LLMs in designing and organizing 3D urban environments from off-the-shelf asset libraries. Such approaches are effective but only create urban environments with existing 3D models, unable to flexibly create new assets when necessary. By comparison, our model can freely create new 3D urban assets in a highly controllable way, allowing for crafting diverse urban environments.

### 2.2 3D WORLD SIMULATOR

A persistent objective in AI research has been to develop machine agents capable of engaging with various environments in 3D space like humans. Toward this goal, researchers have been devoted to building various interactive world simulators in the format of videos (Bruce et al., 2024) or embodied environment (Shen et al., 2022). Existing world simulation environments and platforms are mostly for indoor scenes (Puig et al., 2018; Xia et al., 2018; Kolve et al., 2017; Xia et al., 2020). Differently, Threedworld (Gan et al., 2020) paid attention to creating outdoor environments by retrieving and compositing objects from an existing asset library. When it comes to urban scenes which is the most important open environment, existing works mostly focus on the generative world model for autonomous driving capable of learning scene dynamics and understanding the geometry of the physical world (Hu et al., 2023; Wang et al., 2023b; 2024b). However, these models can only generate new scenes in the format of videos, hard to provide an embodied and interactive urban environment for real use. UGI (Xu et al., 2023) conceptualized some relevant ideas toward urban world simulation, proposing to develop an embodied urban environment for agent development, but still lacks practical implementation. More recently, MetaUrban Wu et al. (2024a) developed a simulation platform for embodied agents in the urban environment, while the provided environment is limited to a fixed style without controllability. To address these challenges, we propose UrbanWorld, which

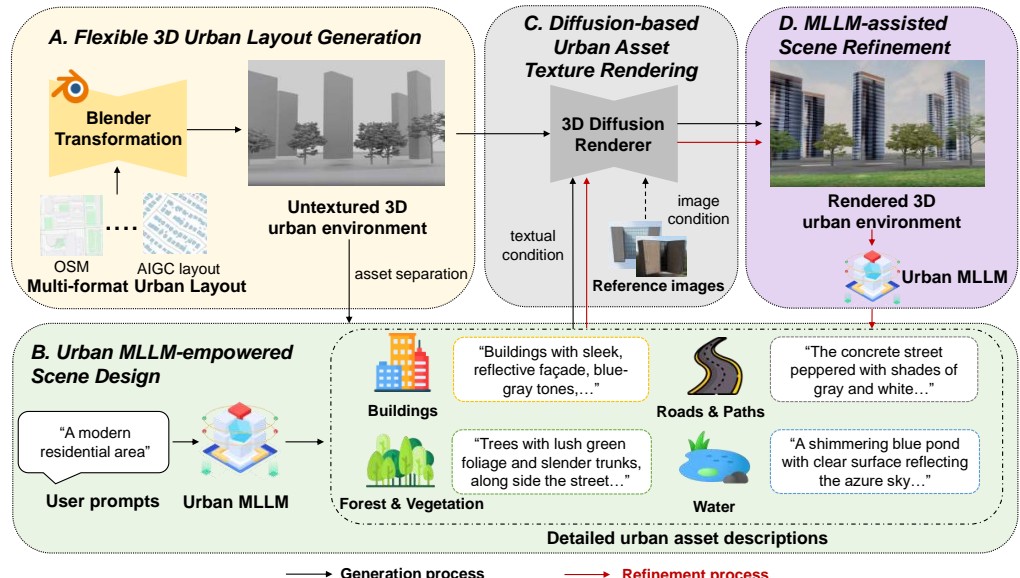

Figure 1: Illustration of the whole framework of UrbanWorld, including four key components: (A) Flexible 3D urban layout generation; (B) Urban MLLM-empowered scene design; (C) Diffusion-based urban asset texture rendering; (D) MLLM-assisted scene refinement.

is expected to facilitate the construction of diverse embodied urban environments with controllable and refined visual appearance, supporting agent development or simulation in various urban scenes.

## 3 METHODOLOGY

There are three main challenges to solve for building a real "urban world model": efficient interactive environment construction, elaborate urban scene planning and high-quality texture rendering. Towards these objectives, UrbanWorld follows a novel "map-design-render-refine" generation pipeline, accomplished by the collaboration of a specialized urban MLLM and an urban asset rendering module. In detail, there are four key stages in UrbanWorld: (1) Flexible 3D urban layout generation, achieving automatic 2.5D-to-3D mapping based on various urban layout metadata such as globally open-accessible OSM data and urban layouts with semantic and height maps, which can address the first challenge. (2) Urban MLLM-empowered scene design, which exploits the superior urban environment understanding ability of a fine-tuned urban MLLM to draft realistic urban scenes emulating human designers for addressing the second challenge. (3) Controllable diffusion-based urban asset texture renderer, achieving flexible urban asset rendering based on 3D diffusion supporting both textual and visual prompts. (4) MLLM-assisted urban scene refinement, exploiting the urban MLLM to conduct reflection to refine the generated urban environment, mimicking the iterative revision in the standard operation process of human designers. The last two components contribute to high-fidelity textures of 3D assets, effectively tackling the third challenge. The overview framework of UrbanWorld is illustrated in Figure 1.

### 3.1 FLEXIBLE 3D URBAN LAYOUT GENERATION

UrbanWorld supports any type of 2.5D layout data as input to efficiently build the untextured 3D urban environment, such as easily accessible OSM data and AI-generated urban layout data. These 2.5D layout data record the metadata of an urban area including the geographic locations and attributes of diverse urban elements such as roads, buildings, vegetation, forests, and water. All urban assets will be automatically separated as independent objects for subsequent element-wise rendering. In this step, UrbanWorld also records the object center location for further reorganization of assets, making it match the real urban layout.

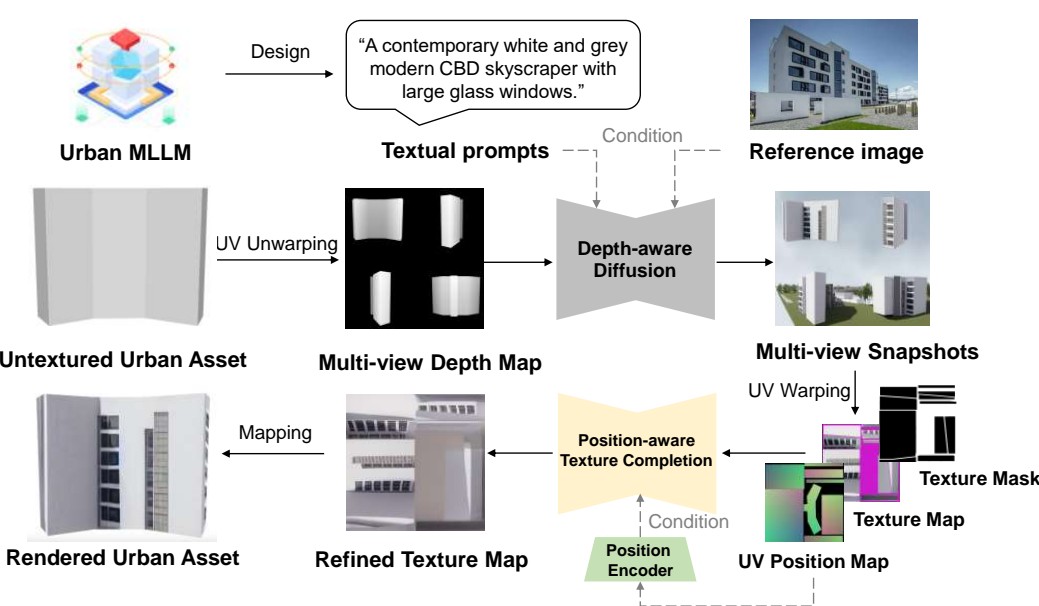

Figure 2: Illustration of the urban asset rendering method in UrbanWorld, mainly including two stages: depth-aware UV texture generation with flexible control under textual and visual prompts and UV position-aware texture refinement.

## 3.2 URBAN MLLM-EMPOWERED SCENE DESIGN

Aiming to effectively craft customized urban environments, UrbanWorld integrates an advanced MLLM fine-tuned on extensive real-world urban imagery data, named Urban MLLM. In detail, we collected approximately 300K urban street-view images from Google Maps and used GPT-4 to generate associated textual descriptions. These descriptions were then manually reviewed and low-quality data were filtered out. Subsequently, we fine-tuned an advanced open-source MLLM, VILA-1.5 (Lin et al., 2024) with the curated dataset to improve its understanding of urban environments. In the workflow of UrbanWorld, Urban MLLM is utilized to act as a human-like designer, which automatically drafts high-quality and detailed urban scene descriptions, ensuring the crafted urban environment are visually coherent and instruction-following. Specifically, taking a simple textual instruction (e.g., "a teaching area in the university") from users as input, UrbanWorld calls Urban MLLM with carefully designed prompts and returns diverse detailed descriptions about visual appearance and materials for each asset. The produced asset descriptions will be used to control the condition of the later texture rendering process.

## 3.3 CONTROLLABLE DIFFUSION-BASED URBAN ASSET TEXTURE RENDERING

Rendering a large-scale urban scene is challenging due to the existence of complex elements and relations, whereby the scene-level rendering will inevitably result in texture mismatching and low-resolution issues. Therefore, we adopt the element-wise rendering strategy to ensure the rendering quality. Simultaneously, in order to speed up the rendering process, we merge some urban element types and finally define four main categories: buildings, roads and paths, forest and vegetation, and water. We implement the rendering with a controllable diffusion-based method consisting of two stages: UV texture generation and texture refinement as shown in Figure 2.

After the 3D environment generation detailed in Section 3.1, we have obtained the untextured 3D mesh of an urban region $S$ and contained assets $\{S_1, S_2, ..., S_i\}$. For each type of asset, UrbanWorld takes the associated textual description $t_i$ from Urban MLLM or a reference image $r_i$ as prompts to control the generation.

We first set a series of camera views $v_i = \{v_i^k\}_{k=1}^N$ to capture multi-view appearances of the asset $S_i$. Next, we utilize the depth-aware ControlNet (Zhang et al., 2023) to control a 2D diffusion model $F$ to generate an image $I_i$ showing the visual appearance of $S_i$ on different views, controlled by the

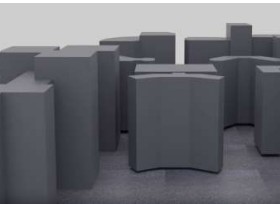 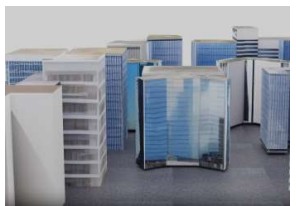 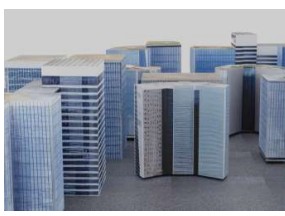

**Untextured Urban Scene**     **Initial Textured Urban Scene**     **Refined Urban Scene**

Figure 3: Illustration of the evolution of created urban environments, including the untextured urban scene, initial textured urban scene and refined urban scene.

condition $c_i \in \{t_i, r_i, (t_i, r_i)\}$:

$$I_i = F(c_i; d_i; z), \tag{1}$$

where $z$ is the latent embedding for the diffusion process, the depth map from different views $d_i$ is obtained from the rendering process $d_i = P(S_i, v_i)$. Then we crop $I_i$ into patches where each represents a unique view $\{I_i^k\}_{k=1}^N$ of the rendered asset. Then we conduct a reverse process $P^{-1}$ of rendering to back-project $I_i$ into the UV texture space:

$$U_i^k = P^{-1}(v_i^k; I_i^k; S_i), \tag{2}$$

Subsequently, we merge the texture maps from different views into a single texture map $U_i$:

$$U_i = \sum_{k=1}^n M_i^k \odot U_i^k, \tag{3}$$

where $M_i^k$ denotes the corresponding mask in the UV space from the view $v_i$.

Up to now, we have obtained the preliminary texture map for the asset $S_i$, while in practice we found that there are still some untextured areas on the object due to the discrete sampling of camera views, especially for assets with many faces. Inspired by the inpainting capability of diffusion models, we introduce an additional UV texture inpainting process to get more complete and natural texture. However, such inpainting can not be directly achieved with general diffusion-based inpainting since the inpainting should be restricted to follow the position relation in the UV texture space. Therefore, we introduce a position map-guided building UV inpainting process, inspired by the inpainting process of general 3D objects (Zeng et al., 2024).

To be specific, we add a UV position map encoder $E_V(\cdot)$ to encode the position map $V_i \in \mathbb{R}^{H \times W \times 3}$, indicating the adjacency relation of the UV texture fragments, where $E_V$ is set to have the same architecture of the image encoder in the diffusion model. Then we curate a set of paired-up UV position maps and UV texture maps for urban assets with complex surfaces and train the position encoder following the pipeline of ControlNet (Zhang et al., 2023). With the control of UV position maps, it's expected to achieve accurate and natural inpainting for UV texture maps. Denote $U_i^*$ as the inpainted UV texture map, the UV inpainting process is formulated as follows:

$$U_i^* = F(c_i; U_i; E_V(V_i)). \tag{4}$$

With the above texture generation and completion, UrbanWorld can produce coherent and high-fidelity textures for various urban elements. For better visual aesthetic of the rendering, we further conduct image upscaling with ControlNet-tile to enhance the structure sharpness and realism of the texture map, contributing to more detailed and realistic appearances of urban assets.

### 3.4 MLLM-ASSISTED URBAN SCENE REFINEMENT

After urban asset rendering, UrbanWorld automatically reorganizes the assets guided by the location information extracted from the metadata of the urban layout, effectively recovering the original urban layout. Inspired by the standard operation process of human designing, where experts will take an overview of the result and conduct iterative adjustments. To mimic such an effort, UrbanWorld resorts to Urban MLLM again to scrutinize the crafted 3D urban environment, especially focusing on

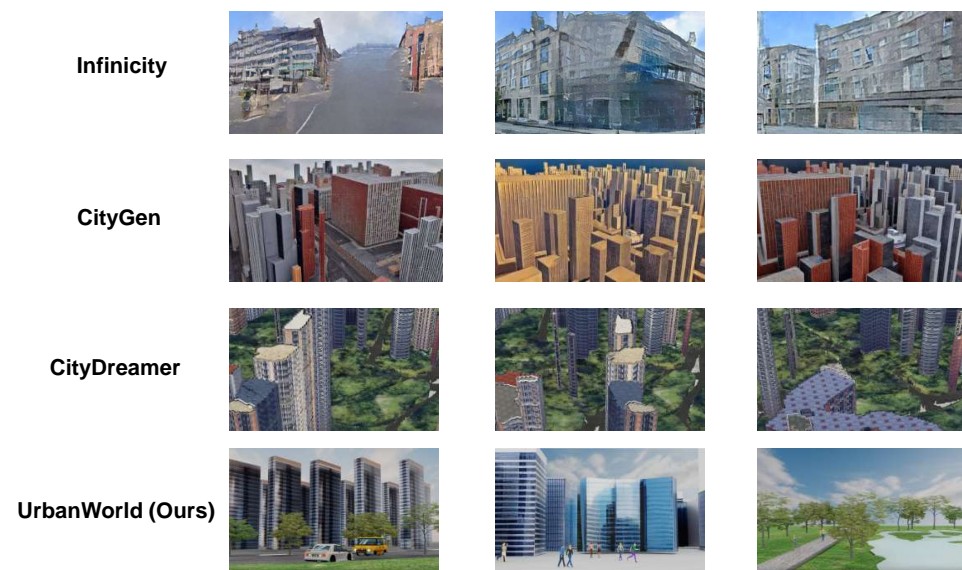

Figure 4: Qualitative comparisons of generated 3D urban environments from Infinicity, CityGen, CityDreamer and UrbanWorld. By comparison, our method can craft more diverse and realistic 3D urban environments enabling dynamic interactions with humans (walking) and vehicles (driving).

the texture details. Specifically, we prompt Urban MLLM to identify the inconsistencies between the generated result and the prompts for generation. Finally, Urban MLLM will provide sophisticated suggestions for further refinement, including elements to be modified and refined design prompts. Then the rendering module described in Section 3.3 will be activated and the involved elements will be rendered under the refined text prompts and updated in the 3D environment. With such a refinement process, UrbanWorld can further align the generated urban environment to user instructions. Here we provide a visualization example in Figure 3, showing the evolution of the created urban environments happened in the running process of UrbanWorld. It can be seen that UrbanWorld works in an iterative refinement manner to create high-fidelity urban environments, where the low-quality textures will be automatically identified and refined with the powerful Urban MLLM.

## 4 EXPERIMENTS

In this section, we first introduce the experimental setup and implementation details (see Section 4.1), and then provide some quantitative evaluations of the created urban environments to demonstrate the superiority of UrbanWorld (see Section 4.2). Finally, we present the generation results of UrbanWorld for qualitative estimation (see Section 4.3).

### 4.1 IMPLEMENTATION DETAILS

UrbanWorld incorporates three key techniques: Blender as the professional 3D modeling software, diffusion-based rendering and Urban MLLM-empowered scene design and refinement. Specifically, we use Blender-3.2.2 for Linux systems and the compatible Blosm addon to handle the OSM transformation. In terms of diffusion-based rendering, we utilize Stable Diffusion-1.5 (Rombach et al., 2022) as the fundamental diffusion backbone, combined with ControlNet-Depth (Zhang et al., 2023) when generating multiple views of 3D assets. We also introduce IP-Adapter (Ye et al., 2023) to support taking reference images as the additional generation condition. We use ControlNet-inpainting (Zhang et al., 2023) as the diffusion controller in the UV texture refinement and ContrlNet-tile (Zhang et al., 2023) in the realness enhancement stage. For the hyper-parameter settings in the rendering part, we set the number of camera views $N = 4$, which can basically satisfy the rendering needs of most urban assets. The number of inference steps in all diffusion processes is set as 30 by default. The UV maps of 3D assets are unwrapped in the "smart projection" mode operated in Blender. All experiments are conducted on a single NVIDIA Tesla A100 GPU.

Table 2: Quantitative evaluation of existing works for 3D urban scene generation and UrbanWorld on depth error, homogeneity index and realistic score.

| Method | FID (↓) | KID (↓) | DE (↓) | HI (↓) | PS (↑) |
|---|---|---|---|---|---|
| SGAM Shen et al. (2022) | 453.81 | 0.522 | 0.575 | 0.872 | 5.6 |
| PersistentNature Chai et al. (2023) | 441.65 | 0.319 | 0.326 | 0.742 | 4.8 |
| SceneDreamer (Chen et al., 2023) | 389.90 | 0.284 | 0.152 | 0.817 | 6.2 |
| CityDreamer (Xie et al., 2024) | 418.38 | 0.210 | 0.147 | 0.830 | 6.0 |
| **UrbanWorld (text)** | 377.65 | 0.187 | 0.089 | 0.683 | 6.5 |
| **UrbanWorld (image)** | **368.72** | **0.154** | **0.082** | **0.665** | **6.7** |

Table 3: Study of the effectiveness of three key designs in UrbanWorld with text prompts as generation conditions: Urban MLLM-empowered scene design, texture enhancement and MLLM-assisted scene refinement.

| Method | FID (↓) | KID (↓) | DE (↓) | HI (↓) | PS (↑) |
|---|---|---|---|---|---|
| UrbanWorld (text) | **377.65** | **0.187** | **0.089** | **0.683** | **6.5** |
| w/o Urban MLLM design | 401.58 | 0.237 | 0.104 | 0.701 | 6.1 |
| w/o texture enhancement | 382.09 | 0.197 | 0.125 | 0.687 | 6.2 |
| w/o scene refinement | 393.73 | 0.202 | 0.096 | 0.690 | 6.3 |

## 4.2 QUANTITATIVE EVALUATIONS

To better demonstrate the superior generation performance of UrbanWorld, in this section, we provide quantitative results on five metrics: Frechét Inception Distance (FID), Kernel Inception Distance (KID), Depth error (DE), Homogeneity index (HI) and Preference score (PS). (more details can be found in Appendix A.1).

We test two versions of our method including UrbanWorld (text) which takes textual prompts for generation and UrbanWorld (image) which takes reference images as generation conditions. The compared methods for urban scene generation include SGAM (Shen et al., 2022), PersistentNature (Chai et al., 2023), SceneDreamer (Deng et al., 2023) and Citydreamer Xie et al. (2024), representing the most advanced performance of automatic 3D urban environment generation. We don't provide results from other methods such as CityGen (Deng et al., 2023), CityCraft (Deng et al., 2024) and SceneCraft (Hu et al., 2024) because the source codes are not open up to now.

From the results presented in Table 2, we can observe that UrbanWorld outperforms on each quantitative metric compared with baselines. UrbanWorld with real street-view images as conditions demonstrates improved performance compared to the version relying solely on textual conditions. In detail, compared with the most competitive baseline, UrbanWorld has 5.4% and 26.7% improvement on FID and KID, indicating better realism of generated results. Besides, UrbanWorld achieves 44.2% improvement on depth error, demonstrating the geometry-preserving ability of UrbanWorld. By comparison, rendering methods such as SceneDreamer and CityDreamer can produce visually appealing urban scenes, but commonly lose geometry consistency. In terms of the homogeneity index, UrbanWorld has 10.4% improvement compared with baselines. Consistent with the observation on the qualitative results in Section 4.3, the generated scenes from existing methods exhibit high homogeneity, limited to the style of training data. By comparison, UrbanWorld can produce more diverse urban environments according to user instruction, effectively achieving customized creation. This makes it possible to craft any type of 3D urban environment that adapts to the needs of different urban environments. Lastly, the created urban environments from UrbanWorld obtain higher preference of GPT-4 outperforming all compared methods on the preference score.

**Effectiveness of three key designs.** To further validate the effectiveness of key designs in UrbanWorld, we conduct ablation studies and show the results on UrbanWorld (text) and UrbanWorld (image) in Table 3 and 4, respectively. We explore the influence of three designs on the performance, including Urban MLLM-empowered urban scene design, texture enhancement and MLLM-assisted scene refinement. The results indicate that all these techniques contribute to the final generation performance. Specifically, the scene design from Urban MLLM contributes most to the quality

Table 4: Study of the effectiveness of three key designs in UrbanWorld with real street-view images as generation conditions: Urban MLLM-empowered scene design, texture enhancement and MLLM-assisted scene refinement.

| Method | FID ($\downarrow$) | KID ($\downarrow$) | DE ($\downarrow$) | HI ($\downarrow$) | PS ($\uparrow$) |
|---|---|---|---|---|---|
| UrbanWorld (image) | **368.72** | **0.154** | **0.082** | **0.665** | **6.7** |
| w/o Urban MLLM design | 373.21 | 0.158 | 0.104 | 0.701 | 6.5 |
| w/o texture enhancement | 384.10 | 0.172 | 0.125 | 0.687 | 6.5 |
| w/o scene refinement | 374.52 | 0.159 | 0.096 | 0.690 | 6.6 |

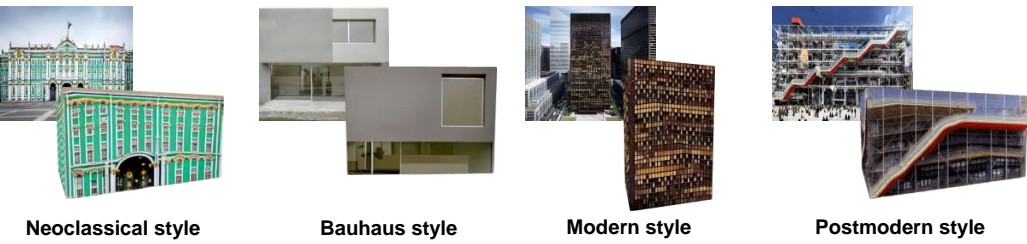

**Neoclassical style**  **Bauhaus style**  **Modern style**  **Postmodern style**

Figure 5: Illustration of the controllable generation of diverse architecture styles when prompting with reference images (upper left) in UrbanWorld.

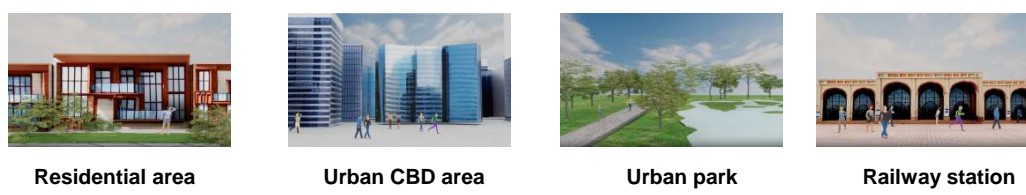

**Residential area**  **Urban CBD area**  **Urban park**  **Railway station**

Figure 6: Illustration of the controllable generation of diverse urban functional scenes guided by different textual prompts in UrbanWorld.

of generated urban environments, benefiting from the rich urban environment knowledge of Urban MLLM. The texture completion and enhancement operation have the most notable effect on the estimated depth error because better texture fidelity can help with geometric perception. Besides, the final scene refinement process leads to a gain in all evaluation metrics, further promoting the generation quality.

### 4.3 QUALITATIVE RESULTS

We present generation results of UrbanWorld produced with textual prompts in Figure 4, including three representative urban functional spaces: residential areas, commercial blocks and parks. For intuitive comparison, we also provide some generation samples from Infinicity (Lin et al., 2023), City-Gen (Deng et al., 2023) and CityDreamer (Xie et al., 2024). The results of Infinicity and CityGen are taken from the original paper because the codes are not open-source. It can be seen that scenes from Infinicity are short of clear textures and well-maintained building structures. Scenes from CityGen feature homogeneous styles without clear characteristics of urban functions. Similarly, the visual appearance of urban elements (especially buildings) in the environments from CityDreamer lacks diversity and is hard to distinguish. Besides, there are also clear geometric distortions of the building boundaries in CityDreamer. These issues will pose great challenges for the real interactions between subjects and urban environments. For example, embodied agents are hard to be trained to conduct urban navigation because the surrounding elements are too similar to recognize. By comparison, the urban elements created by UrbanWorld possess high visual diversity, conveniently

Figure 7: Illustration of the interactive nature of created environments from UrbanWorld, showcasing two representative types of interactions: agent perception and navigation.

controlled by user instructions. Moreover, we explore generating realistic urban assets by prompting UrbanWorld with realistic images as prompts. Figure 5 presents generated urban buildings in various styles guided by realistic images. We can observe that the generated asset possesses high fidelity to the real ones, which is rarely reached by existing approaches. We further extend to generate large-scale urban environments, as illustrated in Figure 6, which presents four representative urban functional spaces, each with a visually consistent and contextually appropriate appearance. These results highlight the superior controllability of UrbanWorld which supports flexible prompts as generation conditions.

## 4.4 INTERACTIVITY DEMONSTRATION

To demonstrate the interactive characteristics of 3D urban environments produced by UrbanWorld, we provide a case study about how UrbanWorld provides information feedback and embodied activity spaces for the agents. We focus on two representative interaction modes between agents and urban environments: perception and navigation, following existing works about urban agents (Wu et al., 2024a; Yang et al., 2024). As shown in Figure 8, the created urban environment can provide multimodal observations for agents including RGB imagery, depth maps and semantic maps. Such information can be utilized to enhance the perception ability of agents in various complex urban environments, benefiting in accomplishing various embodied tasks such as navigation and object manipulation. We also evaluate the agent's navigation capabilities within the generated urban environment, where the task is to navigate to specified target coordinates in 3D environments. Here we utilize the Rapidly-exploring Random Tree (RRT) algorithm for path planning using the start coordinates of agents, target coordinates and obstacle coordinates in 3D space. We record the visual observation trajectory of the agent during navigation as shown in Figure 8, demonstrating the effective interaction between agents and environments.

## 5 CONCLUSION

We propose UrbanWorld, the first generative urban world model to create realistic, customized and interactive 3D urban environments with flexible control conditions in a fully automatic manner. Integrating the powerful urban environment understanding ability of an urban MLLM and the controllable generation ability of diffusion models, UrbanWorld can effectively craft high-fidelity and customized urban environments outperforming existing 3D city generation methods. For practical usage, the created urban environments can provide high-fidelity data and interactive environments for developing embodied intelligence and AI agents. We have contributed UrbanWorld as an open-source tool to benefit broad research communities, which we believe can pave a new way to efficiently establish 3D urban world, accelerating the development of AGI.

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

# A APPENDIX

## A.1 DETAILS OF THE METRICS FOR QUANTITATIVE EVALUATION

**Frechét Inception Distance (FID) and Kernel Inception Distance (KID).** Both metrics quantify the similarity between the distribution of generated images and real images, where lower values indicate better image quality of generated results. We compute FID and KID between frames sampled from the generated scene and an evaluation set comprising 1000 real street-view images randomly sampled from Google Maps. These metrics can effectively measure the realism of generated results.

**Depth error (DE).** Depth error is utilized to evaluate the 3D geometry accuracy, following the implementation of EG3D (Chan et al., 2022) and CityDreamer (Xie et al., 2024). Specifically, we use the pre-trained depth estimation model (Ranftl et al., 2020) to obtain the "ground truth" of depth maps via density accumulation. DE is then calculated as the L2 distance between the normalized predicted depth maps and the "ground truth". The final result is averaged on the result from 100 captured frames of generated urban scenes.

**Homogeneity index (HI).** Realistic cities are featured by complex elements with diverse visual appearances, indicating various functional uses of different urban areas. In order to capture this key character, we propose to evaluate the homogeneity of generated scenes, mainly measuring the variance of different urban scenes. To be specific, we first extract the visual feature of each generated scene image with ResNet (He et al., 2016). The homogeneity index is then calculated as the averaged cosine similarity of each pair of scenes in the feature space. The smaller value of the homogeneity index means a higher diversity of generated urban environments.

**Preference score (PS).** Another widely used approach to evaluate the generated result is utilizing powerful LLMs as the evaluator Wang et al. (2023a); Peng et al. (2024), here we prompt GPT-4 to score the snapshots taken from the generated 3D environment, mainly considering the texture sophistication and geometric completeness (ranging from 1 to 10). A higher score indicates stronger preference for the generated result.

## A.2 DESIGNED PROMPTS FOR URBAN SCENE DESIGN USING URBAN MLLM

```
Urban Scene Generation

cat_name =
    ['building','path_roads','forest','vegetation','water','ground']

user_prompts = "A modern residential area"

instruct_prompts = '''
You are an expert of urban scene design, now I want to generate a 3D
    urban scene of {}, composed of following kinds of assets (building,
    forest, vegetation, water, path_road, ground).
Can you design and generate a caption for each asset (give five types of
    captions for the building) (each caption within 50 tokens), making
    the scene visually look harmonious and realistic.
Only describe the appearance features (must including 'color',
    'material' (such as patterns, roughness, metalness), functional use
    and 'elements' (must include detailed descriptions (color, position)
of the windows, doors for buildings), and don't give too much other
    information. Ensure the texts capable to control text-to-image
    diffusion models like stable-diffusion v1.5.
Please use a dictionary to represent the output result {} without number.
'''

des = ""
for i in range(len(cat_name)):
    des += f"{cat_name[i]}: [Description of {cat_name[i]}]"
    if i < len(cat_name)-1:
        des += ", "
des = "{" + des + "}"

new_prompts = instruct_prompts.format(user_prompts, des)
```

Figure 8: Designed prompt template for urban scene design using Urban MLLM.

