# OpenReview forum: "UrbanWorld: An Urban World Model for 3D City Generation"
_ICLR.cc/2025/Conference — Submitted to ICLR 2025_

### Official Review · Reviewer_tHU8 · 2024-10-19

**Soundness:** 3
**Presentation:** 3
**Contribution:** 2
**Rating:** 5
**Confidence:** 5

**Summary:**

The paper introduces UrbanWorld, a generative model designed to automatically create realistic and interactive 3D urban environments, addressing the challenges of manual labor in urban scene design. UrbanWorld employs a progressive diffusion-based rendering method and a specialized urban multimodal large language model (Urban MLLM) trained on street-view image-text data to guide the generation process. The model consists of four key stages: flexible 3D layout generation, urban scene design using Urban MLLM, controllable asset rendering, and scene refinement. Extensive evaluations demonstrate that UrbanWorld achieves state-of-the-art realism and interactivity, outperforming existing methods like Infinicity and CityGen in generating diverse urban environments suitable for embodied agents.

**Strengths:**

1. The paper is well-written and easy to follow.
2. The code has been released.
3. The overall framework is technically sound. Based on a pre-defined set of common objects in the urban scenario, the framework bridges the gap between the 3D world and 2D views via pre-trained diffusion models. The pipeline is interesting.
4. The framework achieves controllable and customizable scene generation, which can support tasks that require agent-environment interactions.

**Weaknesses:**

1. Even though superior quantitative results are reported, the generated images are not realistic enough based on the demonstration in the paper.
2. It would be better if the authors could provide more diverse qualitative results generated by the proposed method. The proposed system is claimed to be for 3D city generation. It would be good if a sequence of images/video captured with a moving camera is included to show the scene-level generation capability.
3.  I am confused about the UV unwrapping and UV wrapping parts. How can you ensure the wrapping process can align the texture perfectly to the mesh model? For objects of different types and shapes, I believe this process can be hard to model by the diffusion model. The UV unwrapping is usually not unique. Is there any mechanism to enforce the equivariance to different unwrapping manners?
4. I noticed that the Position-awareTexture Completion module is applied to refine the texture map. Can you provide some qualitative results (visualization) to compare the results before and after the refinement?
5. The section 4.4 is a little bit vague. How does your generated environment support navigation? How far is the longest distance your navigation can achieve? It could be better to show a bird-eye-view of your navigation environment.

**Questions:**

Please refer to the weakness part.

---

### Official Review · Reviewer_eF1T · 2024-10-30

**Soundness:** 2
**Presentation:** 2
**Contribution:** 1
**Rating:** 3
**Confidence:** 5

**Summary:**

The paper introduces UrbanWorld, a generative model designed for the automatic creation of realistic, customizable, and interactive 3D urban environments. UrbanWorld employs a progressive, four-stage generation pipeline: flexible 3D layout creation, Urban Multimodal Large Language Model (Urban MLLM)-based scene design, diffusion-based asset rendering, and MLLM-driven scene refinement.

**Strengths:**

UrbanWorld introduces a pipeline that integrates generative diffusion models with an urban-specific MLLM to achieve realistic urban scene creation. This combination allows for controlled generation of 3D assets and adaptive urban design.

**Weaknesses:**

1. The authors claimed section A is flexible urban layout “generation”. However, this is not like generation methods where the distribution of urban layouts are learned from real-world data [1][2][3]. It seems like the authors are just using OSM’s GT data (AIGC-layout is not explained anywhere in the paper). No detail is given on how did the authors transform the OSM data or AIGC data into untextured 3D urban environment. Is there any generation models or other networks involved?  In short, if you are just using GT data and Blender add-on to import it, you can’t call the process “generation”.

2. In section 3.2 and the Appendix A.2, the authors shows a general urban generation prompt is converted into prompts for different categories of urban objects.  However, the same prompt is generated for all objects of the same class. Doesn’t that indicate they would have exact same style and appearance? For example, if there were 50 buildings in the scene, and they all share the same T2I prompt, they end up looking the same. Meanwhile, the authors introduced descriptions for all categories are generated by an MLLM, but did not explain where does the reference image comes from.

3. For a single asset, the authors generated textures from different views conditioned on the same text and reference image, then merged all textures. This approach cannot guarantee consistency between textures as no 3D condition has been used to guide the 2D diffusion model. Meanwhile, it cannot be called “3D diffusion renderer”, since the authors are only inferencing iteratively from pretrained 2D diffusion models.

[1] Infinicity: https://arxiv.org/abs/2301.09637
[2] CityDreamer: https://arxiv.org/abs/2309.00610
[3] CityGen: https://arxiv.org/abs/2312.01508

**Questions:**

1. In Figure 3, texture refinement only shows marginal improvement for buildings. Authors should provide more examples, including other objects and views.

2. In Figure 4, the authors shows existence of human and vehicles, how are these generated?  Are they also assets generated at some stage? Or done by manually post-processing? It is not mentioned anywhere in the paper, and this indicate the visual quality comparison with other methods is completely unfair.

3.Since the framework generate 3D scenes, I suggest the authors to submit videos or at least multi-views of the same scene to demonstrate quality and view consistency of the generated scenes.

---

### Official Review · Reviewer_gfwE · 2024-11-03

**Soundness:** 3
**Presentation:** 3
**Contribution:** 2
**Rating:** 5
**Confidence:** 3

**Summary:**

This paper introduces a method for 3D urban scene creation called UrbanWorld, which facilitates customized and interactive 3D urban world generation. UrbanWorld uses Blender to create untextured 3D layouts from 2D maps and incorporates Urban MLLM to generate textual descriptions for assets. A diffusion-based method is then applied to generate and refine the geometry of the 3D assets.

**Strengths:**

1. The paper is written fluently and is easy to understand.
2. The proposed method shows relatively better results in generating city scenes with assets that have new appearances.
3. The authors effectively showcase various capabilities of the pipeline.

**Weaknesses:**

1. While the authors state that the method achieves “customized, realistic, and interactive 3D urban world generation,” the results appear more simulation-style and fall short of true realism. The texture quality, as seen in Fig. 3 and 4, is not particularly impressive, and there are no significant improvements over CityDreamer.
2. The absence of video results is notable. For a 3D generation task, video demonstrations would better illustrate the quality and realism of the generated scenes.
3. Fig. 4 includes scenes with humans and vehicles, but the method of incorporating these assets is unclear. Details on how these elements are introduced and animated within the scene are missing.
4. Most visual results focus on limited, local areas. For a city-level generation, it would be beneficial to include bird’s-eye-view results covering larger spatial regions, similar to CityDreamer.
5. Including a user study comparison would provide a clearer assessment of the visual quality of the generated scenes.
6. Although the authors claim the ability to create new assets, this appears limited to the level of appearance, with geometry remaining unchanged from the asset library. Given the importance of geometry in 3D generation, this aspect should be addressed.

**Questions:**

Please refer to weaknesses.

---

### Official Review · Reviewer_YwTn · 2024-11-03

**Soundness:** 2
**Presentation:** 3
**Contribution:** 2
**Rating:** 5
**Confidence:** 5

**Summary:**

This paper proposes a generative urban world model that can automatically create a customized, realistic, and interactive 3D urban world with flexible control conditions. The code of this work was released.

**Strengths:**

1. The task of 3D urban generation is important.
2. The method is reasonable and looks to have better quantitative results than previous models.
3. The writing is clear and easy to follow.

**Weaknesses:**

1. Claim of World Model. This work belongs to the 3D urban generation. It is over-claimed to be a world model and barely related to AGI. Authors should precisely identify the task and topic. Then, focus on the specific topic and make it comprehensive rather than claim some large topics.

2. Technical contributions. The motivation of the generation pipeline is unclear. Why do you need a vision language model? What are the special designs in your work different from others, and why do you need them? What is the special challenges that lead you to design the method? So far, the pipeline looks like a combination of recent advanced techniques, i.e., diffusion model and vision language model.

3. Visual results. The visual results are insufficient. From only a few images, it can not be convinced that the visual quality is better than other models. Also, Figure 6 and 4 have some reduplicate results.

4. Evaluation of interactive environments. The evaluation of interactive environments is coarse. The navigation tasks are not really evaluated. From an image, nothing can be evident. What are the quantitative results, and what are the video results? How do you make the simulation of physics? What is the physics engine? What is the training speed? What is the model, RL or IL? What are the evaluation metrics?

**Questions:**

See Weakness.

---

### Meta-Review · Area_Chair_e96G · 2024-12-07

**Metareview:**

**Summary**

The paper presents UrbanWorld, a generative world model for creating interactive 3D urban worlds. The generation consists of four stages: 1) generation of untextured 3D layouts based on input 2D layout, 2) using MLLM to generate textual descriptions detailing the appearance of the assets 3) texturing of urban assets using diffusion based on the generated description 4) scene refinement using a MLLM.

**Strengths**

Reviewers noted the following strengths of the work:
1. The task of 3D urban generation with controllable and customizable scene generation is important [YwTn,tHU8]
2. The method seems to be reasonable [YwTn,tHU8]
3. Code is provided [tHU8]
4. Comparisons show the proposed method to generate better results than previous models [YwTn,gfwE,eF1T]
5. Reviewers mostly found the paper to be well-written and easy to follow [tHU8,eF1T,YwTn]

**Weaknesses**

Reviewers were negative on the submission, and noted that some claims are unsubstantiated, with the main weaknesses being:

1. Inaccurate use of terms.  For instance, claims of the "World Model", "AGI" is not appropriate [YwTn].  The proposed frame is also not a generative model that learns the distribution, and the use of term "3D diffusion renderer" is inaccurate [eF1T]
   - Generated worlds does not seem realistic [gfwE, tHU8]
2. Weak evaluation
   - It's hard to determine from the few visual images (and no videos or birds-eye visuals) that the generated world is actually better than prior work [YwTn, gfwE, tHU8]
   - There was no evaluation of whether the generated environments can be interacted with [YwTn]
   - No human evaluation [gfwE]
3. Some aspects are unclear
   - How are humans and vehicles incorporated? [gfwE,eF1T]
   - How are different appearances for different buildings / objects of the same class obtained? [eF1T]
   - Details of UV unwrapping and wrapping [tHU8]
   - Details of how navigation is supported [tHU8]
4. Design decisions and challenges are not clearly motivated or explained [YwTn]

**Recommendation**

As all reviewers were negative on the submission, and there was no author response, the AC recommends reject.

**Additional Comments On Reviewer Discussion:**

There was no author response, and no reviewer discussion.

---

### Decision · Program_Chairs · 2025-01-22

Reject